

# Characterizing gene tree conflict in plastome-inferred phylogenies

Joseph F. Walker[1,*], Nathanael Walker-Hale[2,*], Oscar M. Vargas[3], Drew A. Larson[4] and Gregory W. Stull[5]

[1] Sainsbury Laboratory (SLCU), University of Cambridge, Cambridge, United Kingdom
[2] Department of Plant Sciences, University of Cambridge, Cambridge, Cambridgeshire, United Kingdom
[3] University of California, Santa Cruz, Santa Cruz, United States of America
[4] University of Michigan—Ann Arbor, Ann Arbor, MI, United States of America
[5] Department of Botany, Smithsonian Institution, Washington, United States of America
[*] These authors contributed equally to this work.

## ABSTRACT

Evolutionary relationships among plants have been inferred primarily using chloroplast data. To date, no study has comprehensively examined the plastome for gene tree conflict. Using a broad sampling of angiosperm plastomes, we characterize gene tree conflict among plastid genes at various time scales and explore correlates to conflict (e.g., evolutionary rate, gene length, molecule type). We uncover notable gene tree conflict against a backdrop of largely uninformative genes. We find alignment length and tree length are strong predictors of concordance, and that nucleotides outperform amino acids. Of the most commonly used markers, *matK,* greatly outperforms *rbcL*; however, the rarely used gene *rpoC2* is the top-performing gene in every analysis. We find that *rpoC2* reconstructs angiosperm phylogeny as well as the entire concatenated set of protein-coding chloroplast genes. Our results suggest that longer genes are superior for phylogeny reconstruction. The alleviation of some conflict through the use of nucleotides suggests that stochastic and systematic error is likely the root of most of the observed conflict, but further research on biological conflict within plastome is warranted given documented cases of heteroplasmic recombination. We suggest that researchers should filter genes for topological concordance when performing downstream comparative analyses on phylogenetic data, even when using chloroplast genomes.

## INTRODUCTION

Chloroplast data have been the most prominent source of information for plant phylogenetics, largely due to the ease with which chloroplast genes can be sequenced, assembled, and analyzed (*Palmer, 1985*; *Taberlet et al., 1991*). The majority of broad-scale phylogenetic studies on plants have used chloroplast genes (e.g., *Chase et al., 1993*; *Soltis et al., 2000*; *Soltis et al., 2011*), and the resulting phylogenies have been used for countless comparative studies examining ancestral states, historical biogeography, and other evolutionary patterns. While older studies relied mostly on targeted genes such as

Corresponding authors
Joseph F. Walker, jfw52@cam.ac.uk
Gregory W. Stull, gwstull@gmail.com

*rbcL* and *matK*, recent advances in DNA sequencing have drastically increased the ease and affordability of whole-chloroplast genome (i.e., plastome) sequencing (*Moore et al., 2006*; *Cronn et al., 2008*; *Cronn et al., 2012*; *Stull et al., 2013*; *Uribe-Convers et al., 2014*), increasing the number of studies employing plastome-scale data for phylogenetic and comparative analyses (e.g., *Jansen et al., 2007*; *Moore et al., 2007*; *Moore et al., 2010*; *Ruhfel et al., 2014*; *Stull et al., 2015*; *Gitzendanner et al., 2018*; *Li et al., 2019*).  Nonetheless, the utility of plastid genes, as well as the entire plastome, is ultimately determined by the extent to which they reflect 'true' evolutionary relationships (i.e., the 'species tree') of the lineages in question (*Doyle, 1992*).

Phylogenomic conflict (i.e., the presence of conflicting relationships among gene trees in a genomic dataset) is now recognized as a nearly ubiquitous feature of nuclear phylogenomic studies (*Smith et al., 2015*; *Rokas et al., 2003*). Gene tree conflict is commonly attributed to biological causes such as incomplete lineage sorting, hybridization, and gene duplication and loss (*Maddison, 1997*; *Galtier & Daubin, 2008*; *Smith et al., 2015*; *Walker et al., 2017*; *Vargas, Ortiz & Simpson, 2017*). The genes within the plastome, however, are generally thought to be free of such biological sources of conflict. This is because the plastome is typically uniparentally inherited (maternally in angiosperms, and at times paternally in conifers: *Mogensen, 1996*; *Birky Jr, 1995*) and undergoes a unique form of recombination that is not expected to result in conflicting gene histories within the plastome (*Palmer, 1983*; *Bendich, 2004*; *Walker et al., 2015*).

In angiosperms, nonmaternal inheritance of the chloroplast has been reported (e.g., *Smith, 1989*; *McCauley et al., 2007*), and several surveys of pollen in flowering plants (*Corriveau & Coleman, 1988*; *Zhang, Liu & Sodmergen, 2003*) have documented plastid DNA in up to 18% of the species examined, indicating potential for biparental inheritance and heteroplasmy (i.e., the presence of two or more different plastomes in a single organism, cell, or organelle). Heteroplasmy, which has been documented in multiple angiosperm species (e.g., *Johnson & Palmer, 1988*; *Lee, Blake & Smith, 1988*; *Hansen et al., 2007*; *Carbonell-Caballero et al., 2015*), creates an opportunity for heteroplasmic recombination, which could result in gene tree conflict in plastome-inferred phylogenies. Unlike animal organelles, where there is little evidence for heteroplasmic recombination (*White et al., 2008*), and thus a strong justification for analyzing the mitochondrial genome as a single unit (*Richards et al., 2018*), heteroplasmic recombination (both intra- and interspecific) has been invoked by multiple studies to explain discordance in various plant clades (*Huang et al., 2001*; *Marshall, Newton & Ritland, 2001*; *Erixxon & Oxelman, 2008*; *Bouillé, Senneville & Bousquet, 2011*). However, only a few recent studies have documented clear evidence of this phenomenon (*Sullivan et al., 2017*; *Sancho et al., 2018*). Beyond heteroplasmy, sharing of genes between the chloroplast and nuclear or mitochondrial genomes remains another potential source of biological conflict (*Martin et al., 1998*; *Martin, 2003*; *Stegemann et al., 2003*; *Rice et al., 2013*; *Straub et al., 2013*; *Smith, 2014*), although transfer back to the plastome would be required to generate conflict and this has rarely been found (*Smith, 2014*). Even though biological conflict in the plastome generally seems rare, the full extent of intraplastome

conflict has yet to be evaluated. Quantifying its extent is of high importance given that most studies assume no conflict and treat the plastome as a single locus (*Wolfe & Randle, 2004*).

Aside from biological sources of conflict, there also remain significant potential sources of systematic and stochastic error that could contribute to conflict across the plastome (e.g., *Burleigh & Mathews, 2007a*; *Burleigh & Mathews, 2007b*). Chloroplast data are used at various time scales, and the accumulation of substitutions over long periods of evolutionary time increases the probability of encountering systematic error due to saturation (*Rodríguez-Ezpeleta et al., 2007*; *Philippe et al., 2011*). Conflict has been demonstrated among different functional groups of genes (*Liu et al., 2012*), among different regions of the plastome (*Davis et al., 2013*; *Walker, Zanis & Emery, 2014*), as well as among individual genes (e.g., *Shepherd, Holland & Perrie, 2008*; *Foster, Henwood & Ho, 2018*; *Gonçalves et al., 2019*). The rate of chloroplast evolution as a whole has been examined (and compared with the nuclear and mitochondrial genomes; *Wolfe, Li & Sharp, 1987*), and rate variation within the chloroplast—especially across the three major regions of the genome, i.e., the large single-copy (LSC) region, the small single-copy (SSC) region, and the inverted repeats (IRa, IRb)—has been explored to help determine the markers useful for phylogenetic inference at different time scales (e.g., *Graham & Olmstead, 2000*; *Shaw et al., 2005*; *Shaw et al., 2007*; *Shaw et al., 2014*). However, no study has comprehensively examined gene tree conflict within the plastome to better characterize the extent and sources of conflict, and to identify the plastid genes most concordant with our current understanding of angiosperm phylogeny (e.g., *Soltis et al., 2011*; *Wickett et al., 2014*; *Zeng et al., 2014*; *Gitzendanner et al., 2018*; but see *Logacheva et al., 2007* for a preliminary investigation of the concordance of individual plastid genes with angiosperm phylogeny).

Here we use phylogenomic tools to characterize the extent of conflict among plastid genes as a function of evolutionary rate, rate variation among species, sequence length, and data type (i.e., nucleotides vs. amino acids) at varying time scales across angiosperms. Our results show that plastid genes, when analyzed in combination, agree with many community consensus relationships; however, across plastid genes there is notable gene tree conflict. The number of conflicting genes at each node is often comparable to the number of concordant genes; however, the majority of plastid genes are uninformative for most nodes when considering support and thus unlikely to positively mislead studies. Information content (gene length and molecule type, i.e., nucleotides vs amino acids) and evolutionary rate were positive predictors of concordance, suggesting that most observed gene-tree conflict is a consequence of spurious inferences from insufficient information. However, some nodes across angiosperm phylogeny show numerous strongly supported conflicting genes; this rare event indicates an area for future exploration. This may result from systematic error or could be the product of a biological source of gene-tree conflict (e.g., heteroplasmic recombination, horizontal gene transfer). We also document the performance of individual genes at recapitulating angiosperm phylogeny, finding the seldom-used gene *rpoC2* to outperform commonly used genes (e.g., *rbcL*, *matK*) in all cases, consistent with previous work highlighting the utility of this gene (*Logacheva et al., 2007*). Although as a whole the chloroplast infers a topology similar to currently hypothesized angiosperm relationships, the conflict underlying it can influence branch

lengths, and therefore dating analyses, comparative methods, and other work that relies on branch lengths could benefit from gene filtering procedures. Our results provide insight into the extent and sources of intraplastome conflict.

## MATERIALS AND METHODS

### Data acquisition and sampling

Complete plastome coding data (both nucleotide and amino acid) were downloaded from NCBI for 53 taxa: 51 angiosperm ingroups and two gymnosperm outgroups (*Ginkgo biloba* and *Podocarpus lambertii*; Table S1). Our sampling scheme was designed to capture all major angiosperm lineages (e.g., *Soltis et al., 2011*) thus allowing accurate inference of angiosperm phylogeny, while also including denser sampling for nested clades in Asterales. This allowed us to evaluate the extent of gene tree conflict at different evolutionary levels/time scales, from species-level relationships in *Diplostephium* (Asteraceae) to the ordinal-level relationships defining the backbone of angiosperm phylogeny.

### Data preparation, alignment, and phylogenetic inference

All scripts and data used may be found on GitHub (https://github.com/jfwalker/ ChloroplastPhylogenomics). Orthology was determined based upon the annotations of protein-coding genes on Genbank; this resulted in near-complete gene occupancy apart from instances of gene loss or reported pseudogenization. We did not use non-coding regions, as proper orthology at deep time scales can be difficult to assess due to gene rearrangements and inversions; this is in part why most deep phylogenetic analyses of angiosperms using plastomes have excluded non-coding data (e.g., *Moore et al., 2010*; *Ruhfel et al., 2014*).

The amino acid and nucleotide data were aligned using Fast Statistical Alignment (FSA v1.15.9; *Bradley et al., 2009*) with the default settings for peptide and the setting "–noanchored" for nucleotide. FSA has been shown to be one of the top-performing alignment programs (*Bradley et al., 2009*; *Redelings, 2014*), and does not rely upon a guide tree for sequence alignment, alleviating downstream bias (*Bradley et al., 2009*; *Boyce, Sievers & Higgins, 2015*; *Chatzou et al., 2018*). A maximum likelihood (ML) tree was then inferred for each gene using RAxMLv.8.2.4 (*Stamatakis, 2014*), with the PROTGAMMAAUTO and GTR+G models of evolution used for the amino acid and nucleotide data, respectively. For each dataset, we conducted 200 rapid bootstrap replicates. The alignments were also concatenated into supermatrices and partitioned by gene using the phyx (v0.99) program pxcat (*Brown, Walker & Smith, 2017*).

The nucleotide and amino acid supermatrices were then each used to infer concatenated 'plastome' trees using the GTR+G and the PROTGAMMAAUTO models, respectively, as implemented in RAxML. To complement the model inference performed by RAxML from the AUTO feature, we also used IQ-TREE's (v1.5.4) (*Nguyen et al., 2014*) built-in model selection process (*Kalyaanamoorthy et al., 2017*) on the partitioned data. We did not partition by codon position because the lengths of most of the plastid genes, especially when divided into three smaller partitions, would have been insufficient to inform an evolutionary model.

Recent work has advocated the use of multispecies coalescent methods when analyzing sets of gene trees from the chloroplast (*Gonçalves et al., 2019*). As such we reanalyzed the nucleotide data using the Maximum Quartet Support Species Tree method as implemented in ASTRAL v5.6.3 (*Zhang et al., 2018*) with default settings.

## Reference phylogenies

For the conflict analyses, described below, we created several reference trees against which the gene trees were mapped. Primarily, we used a topology based on *Soltis et al. (2011)* and where possible that of *Wickett et al. (2014)*, and—for species-level relationships within Asteraceae—*Vargas, Ortiz & Simpson (2017)*. We refer to this as our 'accepted tree', or AT (Fig. 1). We also created a reference tree based on the recent plastome phylogeny of *Gitzendanner et al. (2018)*, which we refer to as the 'Gitzendanner tree', or GT. The GT is highly concordant with the AT, but several deep (and commonly contentious) nodes differ, e.g., the placement of Buxales and ordinal relationships in lamiids (here, we only sampled Lamiales, Gentianales, and Solanales). In the latter case, all three possible combinations of lamiids taxa were assessed (discussed further below in Assessment of Supported Conflict). Although it is difficult to determine which of these trees better represents species relationships of angiosperms, they both serve as useful frameworks for revealing conflict among genes. Here, we primarily focus on analyses using the AT, but conflict analyses using the GT were also conducted to ensure that our overall results were consistent across reference trees.

## Analysis of conflict

All gene trees were rooted on the outgroups using the phyx program pxrr (*Brown, Walker & Smith, 2017*), in a ranked fashion ("-r") in the order *Podocarpus lambertii* then *Ginkgo biloba*; this was performed to account for any missing genes in either of the taxa. In all cases the trees were rooted with at least one gymnosperm. Gene tree conflict in the data was identified using the bipartition method as implemented in phyparts v.0.0.1 (*Smith et al., 2015*), with the gene trees from each data set (amino acid and nucleotide) mapped against the AT and GT. The concordance analyses were performed using both a support cutoff (at ≥70% bootstrap support, i.e., moderate support as suggested by *Hillis & Bull, 1993*) and no support cutoff. When the support cutoff is used, any gene tree node with less than 70% bootstrap support is regarded as uninformative for the reference node in question (i.e., it is uncertain whether the gene tree node is in conflict or concordance with the reference node); when no support cutoff is used, the gene tree node is evaluated as conflicting or concordant regardless of the support value. However, in both cases, a gene is considered uninformative if a taxon relevant to a particular node/relationship is missing from the gene dataset. For example, consider the reference tree ((A,B),C); if a gene tree is missing taxon B, but contains the relationship (A,C), then the gene tree is concordant with the node ((A,B),C) but uninformative regarding the node (A,B). See *Smith et al. (2015)* for an in-depth explanation of this method. To examine patterns of concordance and conflict among all gene trees, we calculated the pairwise Robinson-Foulds distance (*Robinson & Foulds, 1981*) (RF) between each gene tree and the AT, concatenated topology,
and ASTRAL tree. We created a network of trees based on RF distance using the R package igraph v1.2.4.1 in R v3.6.1 (*Csardi & Nepusz, 2006*; *R Core Team, 2018*), using trees as nodes, inverse RF distances as edge weights, and the Fruchterman-Reingold algorithm to lay out the graph (*Fruchterman & Reingold, 1991*).

Levels of conflict were examined across the entire tree and within particular time intervals of the evolutionary history of angiosperms. For the latter case, the ~150 Ma of crown angiosperm evolution were divided into five time-intervals (30 Ma each): 150 to 121 mya, 120 to 91 mya, 90 to 61 mya, 60 to 31 mya, and 30 to 0 mya. The nodes across the tree were binned into the time intervals based on their inferred ages (ages >20 Ma from *Magallón et al., 2015*; ages <20 Ma from *Vargas, Ortiz & Simpson, 2017*; *Roquet et al., 2009*). At each time interval, the proportion of concordant nodes for each gene was calculated (for all the nodes falling within that time interval). This allowed us to assess the level(s) of divergence at which each gene is most informative. We also determined concordance levels of the 'plastome' trees (against the reference) within each of these time intervals.

## Assessments of individual plastid genes

Using the phyx program pxlstr (*Brown, Walker & Smith, 2017*), we calculated summary statistics for each gene alignment and corresponding gene tree: number of included species (as gene alignments could be missing a species from pseudogenization or gene loss), alignment length, tree length (a measure of gene evolutionary rate), and root-to-tip variance (a measure of rate variation across the phylogeny). Alignment length and tree length represent complementary measures of a gene's information content. All else being equal, longer genes should harbor more phylogenetic signal. Tree length serves as a model-informed metric because all sites, and not just those which are variable or parsimony informative, are used in estimation of model distances. Because our study uses maximum likelihood methods, we chose this model-informed metric as opposed to aforementioned metrics. Levels of concordance of each gene tree with the reference trees were then assessed by tabulating the number of nodes concordant between the gene tree and the given reference tree (e.g., the AT). The number of concordant nodes (in Fig. 2 treated as a proportion of total nodes available to support and in Fig. 3 based on total nodes) was used as a measure of a gene's ability to accurately reconstruct the angiosperm phylogeny.

## Predictors of concordance

We examined the statistical relationships between gene tree concordance (using concordance data from the AT-based conflict analysis) and alignment length, tree length, and root-to-tip variance. We treated each node as a trial and analyzed the concordance (success) or conflict (failure) in aggregate, such that each gene was a single observation with $x$ successes in $n$ trials. Because of variable sampling and loss of some genes, $n$ was variable between observations. We analyzed the relationship between probability of concordance and alignment length, tree length, and root-to-tip variance using logistic regression of aggregate binomial trials with the function glm() in R v3.4.4

(*R Core Team, 2018*). Binomial models were generally characterized by high residual deviance, and we thus allowed for overdispersion by fitting quasibinomial logistic regressions (using 'family = quasibinomial()' in R v3.4.4). We fitted models for both amino acid and nucleotide data, both not considering and considering support, and for both alignment methods. We also fitted models for amino acid data inferred only under WAG.

We modeled gene performance as a function of length, tree length, and root-to-tip variation, and as a function of each predictor individually. Because it is possible that apparent relationships between alignment length and concordance may reflect signal from gene information content per alignment site, we also modeled gene performance as a function of length and tree length (as a proxy of gene information content and rate, respectively, as noted above), to assess the relationship between alignment length and gene performance after controlling for variation associated with rate.

Investigation of model fits on full datasets by studentized residuals, leverage, and Cook's distance values revealed that several observations were highly influential. Therefore, we also conducted analyses on reduced datasets to investigate the influence of these observations. In the amino acid datasets, we excluded *rpl22* and *rpl32*, which were probably influential due to their high tree length values, and *ycf1* and *ycf2*, which were probably influential based on their long alignments. In the nucleotide datasets, we excluded *clpP*, which had poor performance relative to tree and alignment length, *rps15*, which had a relatively long tree length, and *ycf1* and *ycf2* for the same reasons as previously noted. Combined analyses of alignment length and tree length were not subject to influence driven by high root-to-tip variance values, and hence reduced datasets had fewer genes removed. In this case, we excluded only *ycf1* and *ycf2*. Regression results were summarized in tables using the R package Stargazer (*Hlavac, 2018*).

## Saturation analyses

We also performed saturation analyses on all the chloroplast genes to determine if they were capable of inferring deep divergences (*Philippe et al., 1994*). In this analysis, pairwise distances between sequences are compared to model-corrected distances as a means of identifying if the real genetic distances may be properly estimated. A saturation analysis was performed on each gene and each codon position and thus the data were realigned by codons for this analysis. The amino acid alignments were used to guide the codon alignments using the program pxaa2cdn, from the phyx package. Four of the genes did not match their corresponding amino acid sequence in its entirety and these were left out of this analysis, since the codons could not be properly aligned. This discrepancy is likely due to errors in the GenBank submission (the amino acid and nucleotide data do not perfectly correspond). Saturation was assessed by determining the observed number of differences between sequences compared to the inferred number of substitutions. This analysis was performed using the "dist.dna" and "dist.corrected" functions in the R package "ape" (*Paradis & Schliep, 2018*), with the F84 model of evolution (*Felsenstein & Churchill, 1996*) used for the correction.

## Comparison of genomic regions

We assessed the utility of the three major plastome regions—the Large Single Copy (LSC) region, Small Single Copy (SSC) region, and the Inverted Repeat (IR) region—for reconstructing angiosperm phylogeny in two ways. First, we constructed ML phylogenies by concatenating all genes found in each region (as described above for each of the plastome trees inferred using all genes). We then calculated the number of concordant nodes with the AT for each genomic region. Second, using the concordance levels of each individual gene (described above), we created a plastome diagram (with genes arranged according to their genomic position) showing the concordance levels of each gene at the five different time scales discussed above (Fig. S1); this permits a qualitative visual assessment of the general concordance levels of each genomic region at each time slice.

## Assessment of supported conflict

We compared four contentious regions of the AT to a series of alternative relationships. The tested alternative relationships were present in either the GT (i.e., the plastome phylogeny of *Gitzendanner et al., 2018*), our plastome phylogenies (with ≥70 BS for the node in question; we refer to this tree below as the 'plastome tree', or PT), and/or three or more of the gene trees from our nucleotide dataset (with ≥70 BS for the node in question). These alternative relationships pertain to: (1) the placements of *Buxus* and *Trochodendron*; (2) relationships among *Acnistus* (Solanales), *Coffea* (Gentianales), and *Olea* (Lamiales); (3) the placements of *Jacobaea* and *Artemisia* in Asteraceae; and (4) the placements of *Galinsoga*, *Ageratina*, and *Parthenium* in Asteraceae. To test these alternative relationships, we used a modified version of the Maxmimum Gene Wise Edge (MGWE) method (*Walker, Brown & Smith, 2018*), where instead of a defined "TREE SET", we used a constraint tree for each alternative hypothesis. The likelihood for every gene was calculated across each constraint, thereby forcing the relationship in question to be the same. Similar to MGWE, this allowed for conflict across the rest of the topology; however, this provides a greater amount of tree space to be explored (similar to *Smith et al., 2018*). This method creates likelihood scores solely based on each alternative relationship, making it robust to gene tree conflict outside of the node in question.

The modification of the MGWE method has been implemented in the EdgeTest.py program of the package PHylogenetic Analysis Into Lineages (https://github.com/jfwalker/PHAIL).

## Assessment of influence of alignment method

Because initial homology statements at the alignment stage can strongly influence the outcome of phylogenetic inference (*Morrison, 2009*; *Morrison, Morgan & Kelchner, 2015*), we repeated the conflict analysis using the same parameters as above with two alternate sets of alignments. In one case we re-inferred using MAFFTv2.271 (*Katoh & Standley, 2013*) with the setting "–auto –max_iterate 1,000". In the second case we cleaned the original FSA gene alignments using pxclsq (*Brown, Walker & Smith, 2017*) for a minimum of 30% column occupancy in order to remove gap-heavy and ambiguously aligned sites, and RAxMLv8 was used.

![PeerJ]

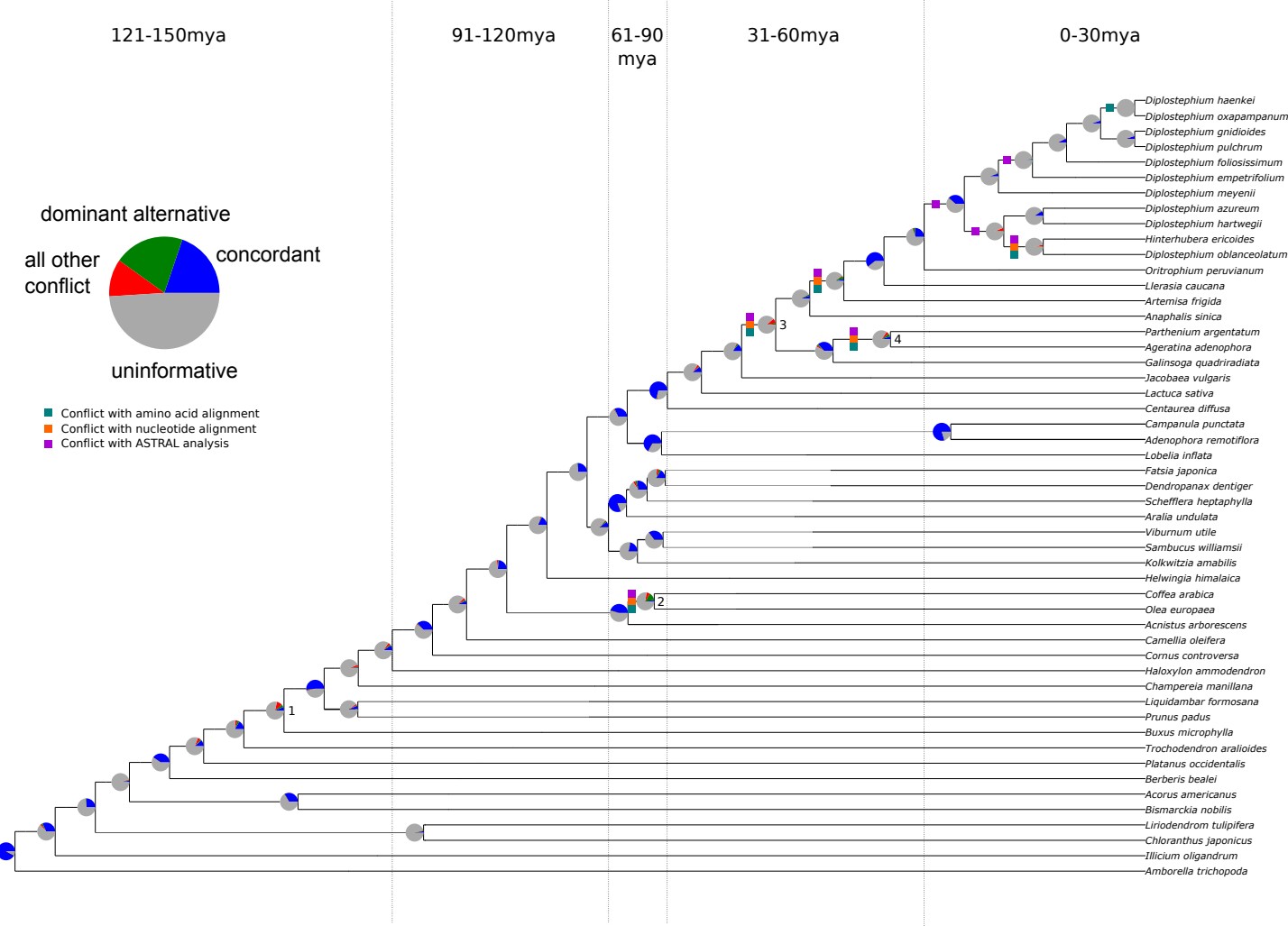

**Figure 1 Summary of chloroplast conflict against the reference phylogeny of angiosperms.** Green, orange, and purple boxes indicate where the amino acid, nucleotide, or MQSST phylogenies conflict with the reference phylogeny. Pie charts depict the amount of gene tree conflict observed in the nucleotide analysis, with the blue, green, red and gray slices representing, respectively, the proportion of gene trees concordant, conflicting (supporting a single main alternative topology), conflicting (supporting various alternative topologies), and uninformative (BS < 70 or missing taxon) at each node in the species tree. The dashed lines represent 30 myr time intervals (positioned based on *Magallón et al., 2015* and *Vargas, Ortiz & Simpson, 2017*) used to bin nodes for examinations of conflict at different levels of divergence.

## Assessment of influence of model

To test if the inferred model of evolution influenced the inference, we re-ran the tree inference using the same methods above on the uncleaned FSA amino acid datasets using only the WAG model of evolution (*Whelan & Goldman, 2001*). The WAG model was not found to be the optimal amino acid model for any gene in the analysis, thus this allowed us to assess the influence of assigning the same sub-optimal model across all partitions.

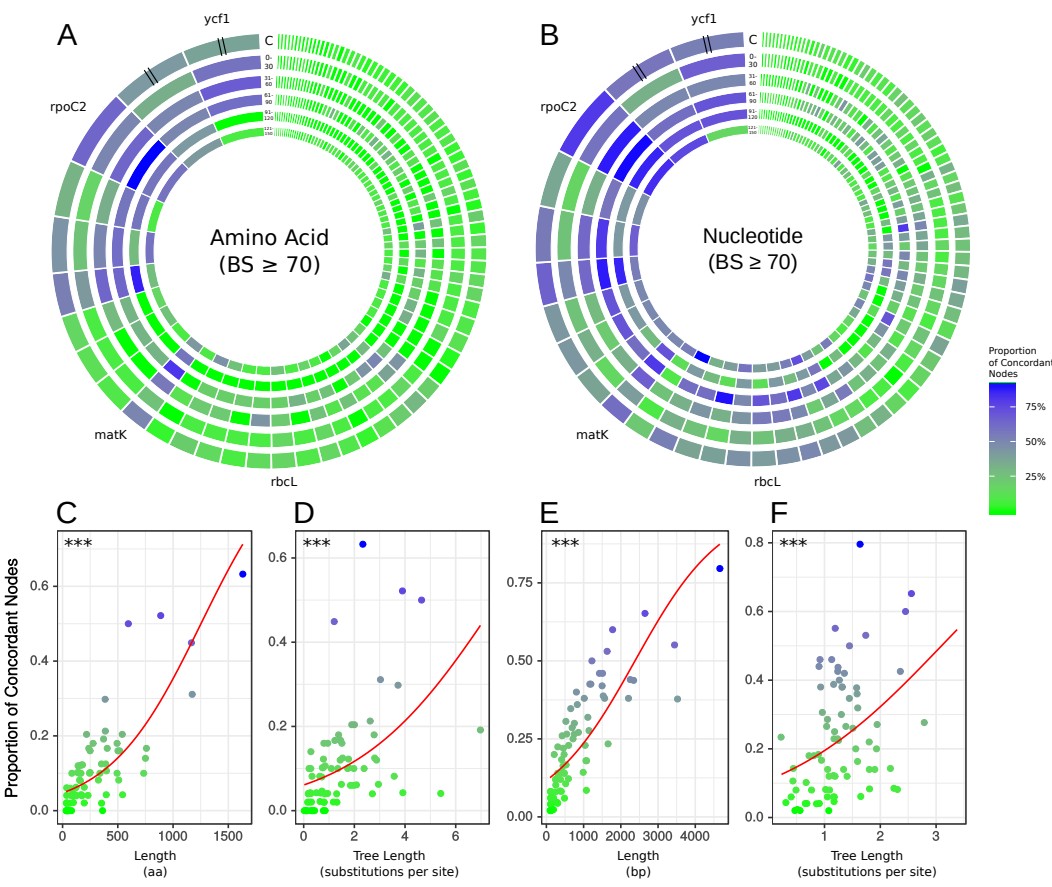

**Figure 2** **Gene tree concordance/conflict at varying time scales.** Each diagram represents a different molecule type and shows the proportion of concordance each gene exhibits at the five time slices shown in Fig. 1: (1) 150–121 mya, (2) 120–91 mya, (3) 90–61 mya, (4) 60–31 mya, (5) 30–0 mya and C is the concordance summed over all the time scales. The individual genes are scaled by length of alignment; however, *ycf1* and *ycf2* are cut to approximately the length of *rpoC2* due to their extremely long alignments. (A) Results from amino acid data considering only nodes with bootstrap support ≥70%. (B) Results from nucleotide data considering only nodes with bootstrap support ≥70%. The plots along the bottom show relationships between gene concordance levels and alignment length and tree length, excluding outlying genes (see Methods). Each point represents the proportion of concordance considering only nodes with bootstrap support ≥70%. Red lines show the predicted values from logistic regression and asterisks give the *p*-value of the relationship from univariate logistic regression, *** = $p < 0.001$. (C) Logistic regression of concordant nodes from amino acid data against alignment length. (D) Logistic regression of concordant nodes from amino acid data against tree length. (E) Logistic regression of concordant nodes from nucleotide data against alignment length. (F) Logistic regression of concordant nodes from nucleotide data against tree length.

# RESULTS

## Influence of alignment and model

The use of MAFFT as opposed to FSA as an alignment tool did not produce a qualitatively different result (Tables S2–S7). Furthermore, our results were largely not qualitatively influenced by using an alignment that had been cleaned for 30% minimum occupancy (Tables S2–S6 and Tables S8–S9). Quasibinomial regression results for amino acids

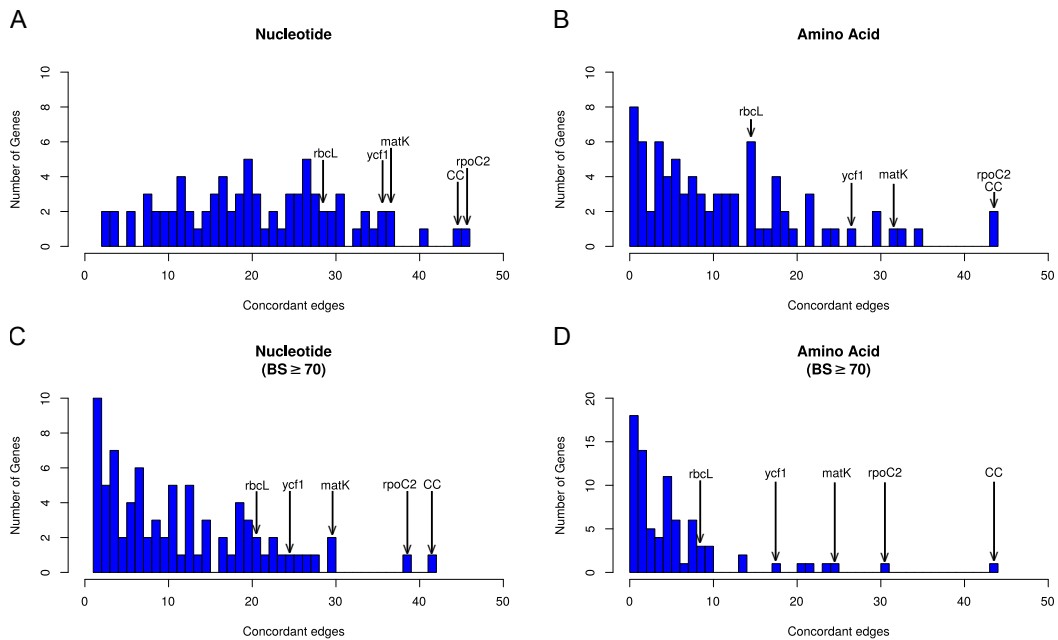

**Figure 3** **Histograms depicting number of concordant edges each gene tree contains compared to the reference phylogeny (i.e., the AT).** Histograms are binned by number of concordant nodes; bar heights give the number of genes in each bin. Commonly used markers (*matK*, *ndhF*, *rbcL*, and *ycf1*) are labeled on the graph, along with the most concordant gene (*rpoC2)* and the number of concordant nodes for the complete chloroplast (CC) compared to the AT. (A) Concordance in the nucleotide dataset not considering bootstrap support. (B) Concordance in the amino acid dataset not considering bootstrap support. (C) Concordance in the nucleotide dataset considering bootstrap support ≥ 70%. (D) Concordance in the amino acid dataset considering bootstrap support ≥70%.

aligned with FSA and cleaned for 30% minimum occupancy were not significant for all predictors, but this model had an abnormally high estimated dispersion parameter (almost 17 times greater than the next largest estimate), suggesting that *p*-values inferred under this model were unreliable (Tables S2–S5). When running a sub-optimal model of evolution for all genes (the WAG model) we recapitulated the results of our model-tested analysis (Tables S2–S5 and Table S10). The differences among all these analyses were occasionally off by one in terms of number of concordant nodes; however, no results qualitatively changed: ranking of genes in terms of concordance, inference from regression models, and detection of strongly-supported conflict were largely unchanged. Therefore, for the remainder of the results and discussion we reference the uncleaned FSA alignment (Tables S11–S12).

## Patterns of conflicting chloroplast signal

The sampling for this experiment allowed us to examine conflict at multiple timescales (the values for each analysis may be found in Tables S6–S12), with roughly 10 divergences occurring in each 30 Ma bin. Our 'plastome' trees, inferred from the concatenated gene sets (nucleotide and amino acid), were highly concordant with the AT (Fig. 1 and Fig. S2. Without considering support, the gene trees showed high levels of conflict across the different analyses. This pattern is similar to that seen in nuclear phylogenomic

datasets (*Salichos & Rokas, 2013*; *Smith et al., 2015*; *Shen, Salichos & Rokas, 2016*). When considering support, the majority of the plastid genes were uninformative for practically all nodes in the phylogeny (Fig. 1; Tables S11–S12); i.e., they had bootstrap support below 70% (moderate support) for that particular relationship (whether in conflict or concordance).

There was no obvious relationship between the amount of gene tree conflict and evolutionary scale (i.e., conflict was relatively evenly distributed across shallow and deeper nodes/time scales; Figs. 1 and 2). Although the greatest degree of gene tree concordance with the AT appeared in the nodes with inferred ages between 90–61 Ma (ages based on *Magallón et al., 2015*), these nodes typically still contained at least 50% uninformative gene trees (Fig. 1). Genome location also showed weak correspondence to conflict/concordance (Fig. S1; Table S13); discussed further below. Rather than timescale and genome location, data type (nucleotide vs amino acid) had a much greater impact on the prevalence of conflict (Figs. 2 and 3), with the amino acid dataset generally showing higher levels of gene tree conflict. When factoring in support (BS ≥70% cutoff), the amino acid data set showed even less concordance with the AT (as more genes were considered uninformative due to low BS support). Considering support also decreased concordance of the nucleotide data set with the AT, but proportionally less. See Figs. S2–S3 for conflict analyses showing the amino acid and nucleotide gene trees mapped onto the AT (the nucleotide results from Fig. S2 are also shown in Fig. 1).

To identify how many different models of amino acid evolution underlie the genes in the plastome, we tested each gene against the candidate set of amino acid models in IQ-TREE and RAxML. We found that a wide range of evolutionary models best fit the individual genes—rather than just a single model that best fits every gene (Table S12). Many of the models were not designed specifically for plastome data, and cpRev, which was designed for plastome data, was only the best fit for 19 of the 79 genes based upon the IQ-TREE model test.

To examine relationships between gene characteristics and levels of concordance/conflict, we calculated the following statistics for each gene: alignment length (a measure of gene information content), tree length (a measure of evolutionary rate), and root-to-tip variance (a measure of variation in evolutionary rate across the tree), all of which can influence properties of phylogenetic inference (*Shen, Salichos & Rokas, 2016*). This information is presented for each gene across both data types in the supplementary information (Tables S11–S12). We used logistic multiple regression to test relationships between concordance and gene characteristics (Tables S11–S21). We found that alignment length had a significant positive multiplicative relationship with odds of concordance across both datasets (Fig. 2, Tables S14–S17). Tree length and root-to-tip variance had significant positive and negative multiplicative relationships, respectively, in both datasets (Tables S14–S15). Notably, excluding highly influential observations did not affect the direction or significance of most relationships (Tables S16–S17), indicating that the results were not due to the influence of these data points. However, this did increase the estimated impact of alignment length on log odds by a factor of 10, probably due to the high leverage exerted by the very long alignments of *ycf1* and *ycf2*. In models including only alignment length and tree length, both predictors had significant positive multiplicative

relationships with odds of concordance even when the other was included in the model (Fig. 2, Tables S18–S21).

We examined patterns of conflict among individual genes by calculating the pairwise RF distance and producing a network graph (Fig. S4). Most genes had large pairwise RF distances to all other genes. We observed a general cluster of genes that were less distant from one another and from the AT and inferred species trees, including *rpoC2* and *ndhF*, that appeared to correlate with alignment length (Fig. S1). In general, the tree set is characterized by high levels of discordance among gene trees as well as between gene trees and the AT and inferred species trees.

## Genomic patterns of concordance/conflict

In terms of number of nodes concordant with the AT, the LSC and SSC regions were roughly comparable, with both outperforming the IR regardless of whether BS support was considered (Table S13). This pattern held across time periods, with the LSC and SSC regions having more concordant nodes than the IR at every time slice. The tree lengths of the LSC and SSC regions (1.82 and 2.05, respectively) were also considerably larger than that of the IR (0.98). The concatenated gene alignment lengths of the LSC , SSC, and IR were 73,422 bp, 10,395 bp, and 19,314 bp, respectively. The genome diagram of concordance (Fig. S1) does not show any striking patterns among the different genomic regions; however, it is notable that the majority of the LSC region is discordant (or uninformative) except for a few highly informative genes (namely, *rpoC2* and *matK* ).

## Performance of individual plastid genes

Across all analyses, *rpoC2* showed the highest levels of concordance with the AT (Fig. 2, Tables S11–S12); in general, when support was not considered, it performed at least as well as the 79 concatenated genes in reconstructing the AT (Fig. 3). The commonly used genes *ndhF* and *matK* generally scored among the best-performing plastid genes (in terms of number of concordant nodes), while *rbcL*, the other most commonly used gene, performed relatively poorly (Fig. 2). The *matK* alignment is ∼250 bp longer than the *rbcL* alignment; the best performing gene, *rpoC2* (alignment length 4,660 bp), is one of the longest plastid genes. However, the notably long region *ycf1*—the unaligned sequence of which is ∼5,400 bp (*Dong et al., 2015*)—did not perform as well as *rpoC2*. In this study, the alignment length of *ycf1* was 21,696 bp (vs. 4,660 bp for *rpoC2*). In several cases *ycf1* performed toward the top; however, it was never the top-performing gene in terms of number of concordant nodes (Tables S11–S12). Despite the high levels of observed conflict overall, we found that every node of the AT was supported by at least one gene. Thus, to varying degrees, all relationships of the AT are found within the plastome gene tree set.

## Saturation analyses

None of the genes analyzed showed significant signatures of saturation (Fig. S5). The uncorrected genetic distances and the corrected genetic distances appeared to be roughly the same, and this trend was true for the first, second, and third codon position. Four genes (*ndhD*, *psbL*, *rpl16*, and *rpl2*) were unable to be properly analyzed for saturation as

**Table 1** Log-Likelihood scores rounded to the nearest whole number for alternative resolutions of contentious nodes based on edge-based analyses using a modification of the Maxmimum Gene Wise Edge (MGWE) method (*Walker, Brown & Smith, 2018*). The best-supported relationship (highest log-likelihood score) for each case is presented in **bold**.

| Contentious relationships | GT | AT | CC | Alternative topology (>3 genes supporting) |
|---|---|---|---|---|
| *Buxus* placement (node 10) | **−618242** | −618371 | −618371 | **−618242** |
| Lamiid relationships (node 19) | **−618203** | −618235 | **−618203** | **−618203** |
| *Jacobaea* and *Artemisia* placements (node 34) | −618535 | **−618506** | N/A (BS <70) | N/A |
| *Galinsoga* placement (node 36) | NA | −618308 | **−618174** | −618319 |

**Table 2** Contentious relationships tested using edge-based analyses. Individual genes supporting, with ≥70, BS the alternative topologies examined for each contentious relationship.

| Contentious relationships | GT | AT | CC | Alternative topology (>3 genes supporting) |
|---|---|---|---|---|
| *Buxus* placement (node 10) | *rbcL, petL, rps2, rps14* | *ndhD, rps4, matK, psbB, rpoC2* | *ndhD, rps4, matK, psbB, rpoC2* | *rbcL, petL, rps2, rps14* |
| Lamiid relationships (node 19) | *ndhK, rbcL, rps11, rps7, ycf4, rpoC2, rps4, ndhA, ndhJ, ccsA, rps2, petB, matK, ndhI, atpB* | *psbB, petA* | *ndhK, rbcL, rps11, rps7, ycf4, rpoC2, rps4, ndhA, ndhJ, ccsA, rps2, petB, matK, ndhI, atpB* | *ndhK, rbcL, rps11, rps7, ycf4, rpoC2, rps4, ndhA, ndhJ, ccsA, rps2, petB, matK, ndhI, atpB* |
| *Jacobaea* and *Artemisia* placements (node 34) | *rpoC2* | None | N/A | N/A |

the nucleotide data uploaded to GenBank were missing the codons for 2 to 3 amino acids in some taxa resulting in an improper codon alignment.

## Analysis of well-supported conflict

We found that, out of the four major contentious relationships in the AT, the edge-based analysis only supports one (Table 1) compared to the alternative hypotheses examined (i.e., the GT, the plastome trees, and the gene trees). In the case of *Buxus* (node 1 in Fig. 1), this analysis supports *Buxus* as sister to Trochodendraceae (Table 1), which is the topology found in the GT and four gene trees: *rbcL, petL, rps2, rps14* (Table 2). In the case of lamiid relationships (node 2), the data support *Olea* (Lamiales) sister to *Acnistus* (Solanales) + *Coffea* (Gentianales), which is the relationship present in the GT and 15 gene trees (Tables 1 and 2). For node 3 (the placements of *Jacobaea* and *Artemisa*), the data support the AT topology, yet no individual gene trees provided ≥70 BS support for this relationship; however, *rpoC2* matched the GT topology with strong support (Tables 1 and 2). Lastly, for node 4 (the placement of *Galinsoga*), the data collectively support *Parthenium argentatum* sister to *Galinsoga* (our plastome tree, PT), with two gene trees (*petA, psaJ*) providing strong support for this topology; however, 16 genes (Table 2) supported a dominant alternative (*Ageratina* sister to *Galinsoga quadriradiata*).

## DISCUSSION

Following the typical assumptions of chloroplast inheritance, we would expect all genes in the plastomes to share the same evolutionary history. We would also expect all plastid genes to show similar patterns of conflict when compared to non-plastid inferred phylogenies. Furthermore, the amino acid and nucleotide plastid data we used should show the same conflicting/concordant relationships against given reference trees (e.g., the AT or GT). Our results, however, discussed below, frequently conflict with these common assumptions about chloroplast inheritance and evolutionary history.

### Conflicting topologies inferred from the chloroplast genome

In general, the 'plastome' topologies inferred from nucleotide and amino acid alignments showed high levels of concordance with the AT (Fig. 1; Tables S11–S12). While the genes within the chloroplast genome are largely uninformative for most nodes of the phylogeny, a number of genes exhibited well-supported conflict (Fig. 1). In general, there appears to be no relationship between evolutionary scale and amount of gene tree conflict (i.e., conflict generally does not appear to correlate with divergence time in angiosperms). Instead, the extent of conflict/concordance had a stronger relationship with the molecule type analyzed. The amino-acid dataset showed the highest levels of gene tree conflict (Fig. 3, Table S12), and the nucleotide dataset had about half the amount of gene tree conflict found in the amino acid data (Fig. 3, Table S11). Support from edge-based analyses (a topology test that is robust to gene tree conflict at nodes other than the one under consideration) for the GT over the AT suggests that at least some signal for conflicting relationships is derived from underlying gene tree conflict when resolving species relationships using all genes. Positive relationships between alignment length, tree length, and concordance also suggest that stochastic error is at play, because shorter or slower-evolving genes harbor less signal to accurately infer relationships. Pairwise comparisons between chloroplast genes indicated that they did not appear to share the same evolutionary history (Fig. S4). Even longer genes were relatively discordant with one another (for example *rpoC2* and *ycf1* had an RF distance of 27). Against a backdrop of largely uninformative genes, long, influential sequences might drive the results of concatenated analyses (*Shen, Salichos & Rokas, 2016*). Indeed, *rpoC2* and *ndhF* had small distances to the AT, concatenated topology, and to one another. Because the AT is a synthetic topology based on analyses of all three plant genomes, we do not consider this to be a potential bias (i.e., the AT reflecting the signal of these genes), but instead indicative of our dataset converging on the signal of the true relationships. In our analysis, however, longer genes still had largely discordant trees.

The superior performance of (coding) nucleotide data compared to amino acid data possibly stems from the relatively greater information content of nucleotides (i.e., longer alignments). Assuming there is not a significant amount of missing/indel data (with the exception of *ycf1* and *ycf2*), longer alignments should result in better-informed models of evolution, all else being equal, aided by both parsimony-informative and -uninformative characters (*Yang, 1998*). However, inherent differences in amino acid and nucleotide models might also explain differences in performance. The nucleotide data were analyzed under the GTR model, where substitution rates among bases are individually estimated; in

contrast, for the empirical amino acid substitution models investigated here, substitution rates are pre-estimated, as the number of estimated parameters for changes among 20 states is extremely large. Additionally, although the plastome has been treated as a single molecule for designing amino acid models of evolution (*Adachi et al., 2000*), a wide variety of amino acid models (some of which were designed for viruses, such as flu or HIV) were inferred to be the best for different plastid genes (Table S12). This might be the result of the different methods implemented in RAxML vs. IQ-TREE for model testing, the different available models (the best-fitting model for some genes in the RAxML analysis was the unpublished matrix DUMMY2), or the lack of sufficient information (because of gene length) to inform the model. However, the most important point is that, based on the amount of information present in each gene, the chloroplast is inferred to evolve under significantly different models of evolution. Given the highly pectinate structure of the AT, phylogenetic inference in this case should rely heavily on the model for the likelihood calculations. While in some cases (e.g., the shortest plastid genes) limited amounts genetic information might be inherently insufficient, in others, improvements in amino acid modeling might lead to great improvements in phylogenetic inference; such has been suggested for animal mitochondrial data (*Richards et al., 2018*). At least insofar as the best-fitting model adequately models the dynamics of sequence evolution, conflict would be expected to decrease relative to a sub-optimal model if systematic error (through inadequate modeling) was the main driver of our results. However, we found that our results were recapitulated when analyzed using only the WAG model, suggesting that this was not the case. These analyses are not exhaustive, because they can only testify to the impact of systematic error resulting from the variation among the model set considered here and do not consider the adequacy of these models. Even though two genes analyzed under two different models produce the same tree, there could still be common assumptions underlying both models that are violated by the dataset, leading to systematic error. Additionally, two genes may share the best-fitting model and produce the same tree, but that model may be misspecified, leading to error. Although not the focus of our study, future examinations of chloroplast phylogenomics could benefit from tests of model adequacy (*Brown, 2014*; *Duchêne, Duchêne & Ho, 2018*; *Chen et al., 2019*).

We expect that at deeper time scales, nucleotides (of coding regions) may begin to experience saturation and thus information loss due to increased noise, at which point amino acids (with 20 states) would begin to outperform nucleotides. However, the time scale of angiosperm evolution does not appear great enough to result in nucleotide saturation (at least for the genes sampled here, given that no genes appeared to exhibit significant levels of saturation; Fig. S5), indicating that nucleotides outperform amino acids for phylogenetic analysis of plastomes across angiosperms. Future work, with a broader plastome sampling across green plants, will be necessary to determine the evolutionary scale at which amino acids become more informative than nucleotides for phylogenetic inference.

Taxon sampling may also affect the accuracy of phylogenetic inference, and it is possible that some of the incongruence observed here is due to the influence of missing taxa (*Zwickl & Hillis, 2002*; *Heath, Hedtke & Hillis, 2008*). There is debate about the relative impact of

more sequences versus more taxa (*Rosenberg & Kumar, 2001*; *Zwickl & Hillis, 2002*; *Rokas & Carroll, 2005*). However, our dataset sampled most major angiosperm lineages, and the total analysis was highly concordant with the AT and the GT. Furthermore, patterns of conflict in this dataset reflect those detected in more targeted sampling efforts (*Foster, Henwood & Ho, 2018*). Taken together, this suggests that taxon sampling did not play a large role in our results.

## Analyses of well-supported conflict

The use of a reference topology (AT) afforded us the ability to examine how individual genes within the plastomes agree/conflict with our current, generally accepted hypothesis of angiosperm phylogeny. However, several contentious areas of angiosperm phylogeny remain unresolved, with different topologies recovered across different analyses (e.g., the AT, GT, and PT show small differences in several parts of the tree). Thus, we used edge-based analyses to compare these alternative topologies.

The placement of *Buxus* (Buxales) is a salient example, with different analyses resolving different placements. Our edge-based analyses found *Buxus* sister to *Trochodendron*/Trochdendrales (i.e., the GT topology) to have the highest likelihood score (Table 1), and this placement was recovered in four plastid genes (*rbcL*, *petL*, *rps2*, *rps14*) with strong support. However, five plastid genes recovered, with strong support, *Buxus* and *Trochodendron* successively sister to the core eudicots (i.e., the AT topology): *ndhD*, *rps4*, *matK*, *psbB*, and *rpoC2*. Relationships among core lamiid orders are another intriguing example of gene tree conflict. Core lamiid relationships have been notoriously problematic (*Refulio-Rodriguez & Olmstead, 2014*; *Stull et al., 2015*), and previous phylogenetic studies have recovered every possible combination of the constituent orders (Boraginales, Gentianales, Lamiales, Solanales). While our sampling does not include Boraginaceae, we tested alternative relationships among the three remaining orders. The relationship (Lamiales(Gentianales,Solanales)), which is present in the GT, received the highest likelihood support and is found in 15 gene trees (Tables 1 and 2). However, the alternatives were each recovered with strong support by two plastid genes: *psaA* and *atpH* (Gentianales sister to the rest), and *psbB* and *petA* (Solanales sister to the rest). Asteraceae also exhibited several examples of strongly supported conflict (Tables 1 and 2), but sampling differences across the trees examined make comparisons more difficult.

It is difficult to determine the causes of the observed instances of strongly supported conflict, which can be found at most nodes in the phylogeny (Fig. 1, Tables S11–S12). Based on the present analyses, we cannot conclude whether this conflict is biological (e.g., genes with different evolutionary histories in a single genome) or a consequence of systematic, stochastic, or other analytical errors (e.g., modeling error, lack of information or misalignment). We have endeavored to control for these factors. For example, it is possible that alignment errors could be responsible for instances of strongly supported conflict (*Richards et al., 2018*). However, amongst genes with strongly supported conflict, no obvious misalignment was present, and recapitulation of the results when testing multiple alignment methods and a cleaning procedure suggests misalignment, if present, does not explain the patterns observed here. Although the findings of *Richards et al.*
*(2018)* are supported by a lack of evidence for recombination in animal mitochondria (*White et al., 2008*), there are reasons to believe that biological gene tree conflict could occur in the chloroplast genome (*Sancho et al., 2018*), but we stress that we make no attempt to fully characterize this. In parallel to our work, *Gonçalves et al. (2019)* have analyzed the prevalence of conflict across the plastome and have advocated for the use of both concatenation and multispecies coalescent methods (MSC) for inferring full plastome relationships. Our own analysis of the data with the summary coalescent method ASTRAL (*Zhang et al., 2018*) demonstrates that the different methods result in different relationships (Fig. 1). Specifically, we found much of the conflict co-occurs with conflict from the full plastome supermatrices, with the exception of the generic level resolutions of *Diplostephium*. We note that a core assumption of the MSC is the lack of linkage amongst loci, likely violated by the chloroplast in all but a small number of cases (Edwards et al., 2016; *Gonçalves et al., 2019*). See *Gonçalves et al. (2019)* for an overview of coalescent methods applied to plastome data. With our data, no analyses we performed fully explained the level of conflict observed and we suggest that the possibility of biological conflict deserves further exploration, especially given the recent documented instances of inter-plastome recombination (*Sullivan et al., 2017*; *Sancho et al., 2018*) and exchange between the chloroplast and other genomes (*Martin et al., 1998*; *Martin, 2003*; *Straub et al., 2013*).

## Implication for chloroplast phylogenomics

These examples above are not meant to represent an exhaustive examination of conflicting relationships within angiosperms. Instead, they are intended to highlight several instances of strongly supported conflict within the plastome. These results nevertheless have important implications for plastid phylogenomics, suggesting that concatenated analyses should be performed with caution given the notable presence of gene-tree conflict. The majority of nodes show at least one strongly supported conflicting gene, and many nodes have roughly equivalent numbers of conflicting and concordant genes against the reference tree (Fig. 1). Because of the mode of inheritance of the chloroplast, it has been common for authors to assume that gene tree incongruence occurs due to stochastic error, and thus perform concatenation analyses on the whole chloroplast to increase phylogenetic signal. One of the key assumptions of concatenation methods is that all genes share the same underlying tree. While the signal of stochastic error (e.g., longer genes being more concordant) supports this approach, our detection of supported conflict suggests that this approach may be less valid than previously assumed. Concatenated analyses with extensive underlying conflict can yield problematic results in terms of both topology and branch lengths (*Brown & Thomson, 2017*; *Shen, Hittinger & Rokas, 2017*; *Walker, Brown & Smith, 2018*). Particularly against a backdrop of largely uninformative genes, a few genes exhibiting conflict due to stochastic error, inadequate modeling, or biological sources can drive the analysis toward the wrong topology (*Brown & Thomson, 2017*; *Shen, Hittinger & Rokas, 2017*; *Walker, Brown & Smith, 2018*). We suggest that authors should proceed with caution when assuming that all chloroplast genes share the same tree.

## Utility of individual plastid genes for future studies

Previous studies have laid a strong framework for determining the utility of chloroplast regions at various phylogenetic scales. For example, work by *Shaw et al. (2005)*, *Shaw et al. (2007)* and *Shaw et al. (2014)* highlighted non-coding DNA regions useful for shallow evolutionary studies, while *Graham & Olmstead (2000)* explored protein-coding genes useful for reconstructing deep relationships in angiosperms. Here, we expand upon previous work by using a novel phylogenomic approach, allowing us examining the concordance of individual protein-coding plastid genes with all nodes of the accepted angiosperm phylogeny (AT; as well as slight variations thereof, e.g., the GT). We paid special attention to *matK* and *rbcL*, given their historical significance for plant systematics (e.g., *Donoghue et al., 1992*; *Chase et al., 1993*; *Hilu et al., 2003*). We find that *rbcL* performs relatively well in recapitulating the AT—however, *matK* performs considerably better (i.e., it generally has more nodes concordant with the AT; Tables S11–S12). This is likely due to a strong positive correlation between alignment length and number of concordant nodes, as noted above (Fig. 2); *matK* has a longer alignment/gene length than *rbcL*.

The gene *ycf1* has been found to be a useful marker in phylogenetics (e.g., *Neubig et al., 2009*; *Neubig & Abbott, 2010*; *Thomson, Vargas & Dick, 2018*) and barcoding (*Dong et al., 2015*), and here we find that it generally performs above average. The alignment of *ycf1* is abnormally long and contains a large number of gaps, and this is likely due to its position spanning the boundary of the IR and the SSC, an area known to fluctuate greatly in size. This variability likely contributes to the value of *ycf1* as a marker for 'species-level' phylogenetics and barcoding. However, the performance of *ycf1* does not scale with its alignment length. In terms of concordance, we find that *matK* performs roughly equally as well if not slightly better than *ycf1* (Tables S11–S12). This might in part be a consequence of *ycf1* being missing/lacking annotation from some species, preventing us from analyzing its concordance/conflict with certain nodes. Nevertheless, our results add to the body of evidence supporting *ycf1* as a generally useful plastid region. However, we found *rpoC2* to outperform all other plastid regions in every case (Tables S11–S12), and its alignment length (4,660 bp) is ∼1/5th the length of *ycf1*, easing the computational burden of using alignment tools such as FSA and BAli-Phy (which would struggle with a region as long as *ycf1*), as well as divergence dating and tree-building programs such as BEAST (*Suchard & Redelings, 2006*; *Drummond & Rambaut, 2007*).

In our analyses, when BS support is not considered, *rpoC2* performed at least as well if not better than using the concatenation of all chloroplast genes (Fig. 3; Tables S11–S12). When support is considered (Fig. 3; Tables S11–S12), *rpoC2* still remains the best-performing gene, but it performs slightly worse than the concatenation of all chloroplast genes (in terms of number of supported nodes concordant with the AT). The utility of *rpoC2* likely stems from its notable length, resulting in a wealth of useful phylogenetic information. Biologically, even long genes may be functionally constrained such that not all sites are variable, such as *rbcL*, which performed relatively poorly here (*Kellogg & Juliano, 1997*). However, were this to be the case for *rpoC2*, its longer length relative to *rbcL* should be expected to counter this effect. In light of our results, *rpoC2* should be a highly attractive coding region for future studies, as it generally recapitulates the plastome phylogeny while allowing more proper

branch length inferences (given that conflicting signal among multiple genes can result in problematic branch length estimates: *Mendes & Hahn, 2016*). Recent work (*Salichos & Rokas, 2013*; *Smith, Brown & Walker, 2018*; *Steenwyk et al., 2018*) indicates that filtering for a smaller set of highly informative genes might yield more accurate results in various phylogenetic applications (e.g., topology estimation, divergence dating). Given this, *rpoC2* might be particularly useful for comparative analyses requiring accurate branch length estimates (i.e., the majority of comparative analyses). Use of *rpoC2* alone (instead of the entire plastome) would also allow for more complex, computationally expensive models to be implemented. Furthermore, focused sequencing of *rpoC2* would increase compatibility of datasets from different studies, facilitating subsequent comprehensive, synthetic analyses. Although the performance *rpoC2* may be dataset dependent, our results support its utility at multiple levels, in terms of both time scales and sampling. It is important to note, however, that we did not include non-coding regions in our study, which would likely outperform coding regions at shallow phylogenetic levels. As previously noted, however, it is generally too difficult to use non-coding regions across broad phylogenetic scales (e.g., angiosperms) because of complications related to homology assessment and alignment.

## Genomic patterns of concordance/conflict

Several previous studies (e.g., *Jian et al., 2008*; *Moore et al., 2011*) have highlighted the Inverted Repeat (IR) as a valuable plastid region for deep-level phylogenetic analyses, attributing its utility to its relatively slow rate of evolution, resulting in less homoplasy and minimal saturation. Our results contradict this notion, suggesting that the coding sequences of the IR alone perform poorly compared to the LSC and SSC coding regions for reconstructing angiosperm phylogeny (Fig. S1 and Table S13). It is important to note that there are important differences between our study and earlier studies on the IR (e.g., *Jian et al., 2008*; *Moore et al., 2011*). For one, we did not include the ribosomal RNA genes, which are highly conserved; thus if the conserved nature of the IR (or at least portions of it) is the basis of its utility, then this might explain the poor performance in the current study. Our saturation analyses, however, (described above) did not reveal any genes to have significant saturation issues at the scale of angiosperm evolution. This calls into question the idea that the conserved genes of the IR would make it superior for reconstruction of angiosperm phylogeny. Instead, it is possible that the non-coding regions of the IR (which we did not include here) are highly informative for angiosperm phylogeny. While the non-coding regions of the LSC and SSC regions have been extensively examined for use as phylogenetic markers (*Shaw et al., 2005*; *Shaw et al., 2007*; *Shaw et al., 2014*), the non-coding regions of the IR have been underexplored. Among the IR genes examined here, *ycf2* (which is exceptionally long) showed the greatest levels of concordance (Fig. S1 and Tables S11–S12), underscoring the idea that longer genes are generally more useful for phylogeny reconstruction.

Another important difference between our study and *Moore et al. (2011)* is that we only partitioned our data by gene region, while *Moore et al. (2011)* explored various partitioning strategies (including codon positions and different combinations of genes). It is clear that sequences within the plastomes follow various different models of molecular evolution (as

shown above in the results section *Patterns of conflicting chloroplast signal*). Exploration of more complicated partitioning schemes—which is a time-consuming process (*Lanfear et al., 2012*; *Kainer & Lanfear, 2015*), and beyond the scope of this study—might generally improve plastome-inferred phylogenies.

## Conclusions

We find notable levels of gene-tree conflict within the plastome, at all levels of angiosperm phylogeny, highlighting the necessity of future research into the causes of plastome conflict: do some genes share different evolutionary histories, or is systematic error (e.g., poor modeling) the source of the observed conflict? With rare exceptions, most genes are largely uninformative and, when analyzed as a whole, are therefore unlikely to mislead researchers. However, our results call into question the appropriateness of concatenating plastid genes for phylogeny reconstruction without some level of scrutiny. Even if the concatenated topology is largely congruent with the 'true' species tree of a given group, underlying conflict can mislead branch length estimation (*Mendes & Hahn, 2016*). Therefore, we recommend that comparative methods and/or dating analyses that rely on branch lengths use filtering procedures similar to those proposed for nuclear data (*Salichos & Rokas, 2013*; *Smith, Brown & Walker, 2018*; *Steenwyk et al., 2018*).

We find alignment length and molecule type (nucleotide vs amino acids) to be the strongest correlate with plastid gene performance—i.e., longer genes and nucleotides (which have greater information content than amino acids) generally show higher levels of concordance with our accepted hypothesis of angiosperm phylogeny. This also shows that as a single unit, *matK*, *rbcL* and the other commonly used marker *ndhF* are not up to the task alone and if possible should be replaced by longer genes such as *rpoC2 , ycf1, and ycf2*. Our findings support the notion that longer genes (with greater information content) are generally superior for phylogeny reconstruction, in line with theoretical predictions (*Yang, 1998*) and should be prioritized when developing phylogenetic studies.

## ACKNOWLEDGEMENTS

We thank S.A. Smith and M.J. Moore for reading an early draft of the manuscript and proving helpful suggestions toward its improvement. We also thank A. Alverson, Liming Cai, two anonymous reviewers and S. Edwards for helpful comments that improved the manuscript.

### Funding

Joseph F. Walker was supported by a Rackham Predoctoral Fellowship. Gregory W. Stull was supported by a NSF Postdoctoral Fellowship (NSF DBI grant 1612032). Oscar M. Vargas was supported by NSF FESD 1338694 and DEB 1240869. Drew A. Larson was supported by NSF DEB grant 1458466. Nathanael Walker-Hale was supported by a Woolf Fisher Cambridge Scholarship. The funders had no role in study design, data collection and analysis, decision to publish, or preparation of the manuscript.

## Grant Disclosures

The following grant information was disclosed by the authors:
NSF Postdoctoral Fellowship: 1612032.
NSF FESD: 1338694.
DEB: 1240869.
NSF DEB: 1458466.
Woolf Fisher Cambridge Scholarship.

## Competing Interests

The authors declare there are no competing interests.

## Author Contributions

- Joseph F. Walker and Nathanael Walker-Hale conceived and designed the experiments, performed the experiments, analyzed the data, contributed reagents/materials/analysis tools, prepared figures and/or tables, authored or reviewed drafts of the paper, approved the final draft.
- Oscar M. Vargas and Drew A. Larson conceived and designed the experiments, analyzed the data, approved the final draft.
- Gregory W. Stull conceived and designed the experiments, analyzed the data, authored or reviewed drafts of the paper, approved the final draft.

## Data Availability

Data and code are available at GitHub: https://github.com/jfwalker/ChloroplastPhylogenomics.

## Supplemental Information

Supplemental information for this article can be found online at http://dx.doi.org/10.7717/peerj.7747#supplemental-information.

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
