# Peer review of "Characterizing gene tree conflict in plastome-inferred phylogenies"

_PeerJ, doi:10.7717/peerj.7747_

## Round 0.1 · original submission · Minor Revisions

Thank you for submitting this interesting paper. In addition to the extensive (although relatively minor) comments by the reviewers, I would like to suggest a few revisions of my own, especially since the approaches and assumptions of the paper are quite different from those of animal biologists. I wonder if you could insert a few additional sentences in the introduction documenting the direct evidence for recombination in cpDNA outside of gene tree conflict. Is there direct evidence for recombination in cpDNA from genetic studies? I ask because recombination in animal organelle genomes is highly controversial and still not resolved, so it was a surprise to me that cpDNA is widely assumed to undergo recombination among systematists. But of course we can't use the variation in gene trees themselves as evidence of recombination, although of course it is consistent with that. If there is no direct evidence for recombination other than gene tree conflict, it might be worth stating this directly, and perhaps even alluding to the assumed lack of recombination in animal mtDNA (as indicated by papers like Richards et al. Syst. Biol. 67(5):847–860, 2018)

I would also agree with the suggestion of reviewer 2 that you need to analyze your data with a phylogenetic method that incorporates gene tree variation, such as a coalescent method. But it is important to do that while being transparent about your assumption of recombination, since recombination between loci is a requirement for using such methods. The gene trees don't have to be different but they do need to be genetically independent. Without meaning to push my own work, you can read more about these issues in Edwards SV. 2009. Evolution, 63:1-19.

·

Basic reporting

Walker and Walker-Hale et al. characterized various aspects of gene tree conflicts among plastid coding genes, highlighting the phylogenetic utility of long genes, especially rpoC2. The study is carefully designed, analyses are thorough, and conclusions are reasonable. The figures are informative and easy to understand. The manuscript is well-written in unambiguous language.
In the introduction, the authors summarized biotic and abiotic sources of conflicting phylogenetic signals in molecular datasets, including a paragraph reviewing plastid gene tree conflict due to biparental inheritance and recombination (line 65-83). Yet this study is focused on investigating the abiotic origins of gene tree conflicts. As concluded in the study, analytical error is the primary source of observed gene tree conflicts and the authors do not specifically test for hypothesis of horizontal processes, except that some strongly conflicting genes suggest such possibilities. As a result, these contents should be greatly reduced in the introduction or moved to discussion where it is more appropriate to be presented.
In addition, the authors should also make the reference format consistent throughout. For example, line 680-681: the publication year should follow author names and ‘Algorithms for molecular biology’ should be capitalized. In many cases, the publication year are in parenthesis, which is not consistent with PeerJ standard reference format.

Experimental design

Gene tree conflict among organelle genes is a well-recognized phenomenon. The authors examined how well phylogenetic conflict can be predicted by tree length, sequence length, and data type. Such conflict is measured by comparing phylogenies to two reference trees—tree AT from Soltis et al. (2011) and tree GT from Gitzendanner et al. (2018). The GT tree is built entirely on plastome and eight out of the 17 markers used in the AT tree are also plastid. Using these two trees as reference tree may potentially cause some bias in assessing conflicting phylogenetic signals since we expect plastid phylogenies recovered in this study to match these two trees. The ‘good’ genes identified in this study may be the genes that contain strong phylogenetic signal and drive the AT or GT trees rather than the genes that reflect the true evolutionary history (as discussed in line 544-550). In addition, the AT and GT trees are built using concatenation method without dealing with the issues of ‘strongly conflicting’ signals among plastid genes as identified in this study. I think the authors should at least include in discussion how do the reference trees compared to recent multi-locus nuclear phylogeny based on coalescent methods (e.g., Wickett et al. 2014 and nuclear species tree from the oneKP project deposited at github https://github.com/smirarab/1kp).

The phylogeny is stratified in order to assess phylogenetic utility at both deep and shallow levels. Yet they seem to be only used for visualization purposes in the figures. When discussing the performance of individual plastid genes (line 383-396), each individual gene was assessed based on overall concordance across the entire tree and all evolutionary depths. Such way of summarization cannot fully reflect the benefit of stratified analysis. Thus, in addition to the current analyses, I suggest the authors assessing the performance of genes as well as different genomic regions (LSC, SSC, and IR) separately in the five time windows to reveal any more detailed patterns. It is also surprising that IR does not perform as well as the other two genomic regions at deep phylogenetic scales. This may be a potential bias because the authors used the number of concordant/discordant genes. This result can be influenced by the length and number of genes in that genomic regions especially for smaller genomic regions as SSC and IR. The utility of genomic regions may be better assessed as proportion of concordant sites ranger than genes.

Validity of the findings

The conclusions are generally well supported by the evidence provided in the study. The authors particularly highlighted the highly informative gene rpoC2 as a marker for phylogenetic reconstruction. In line 119-122, the authors suggested that analyses that relies on branch length should also benefit from rpoC2 for accurate topology estimation. I don’t think the author should make such suggestions because accurate branch length estimation is much harder than getting the topology right. Whether rpoC2 yield a more accurate branch length is not tested in the study, especially compared to a concatenated matrix of various genes.

Additional comments

Minor comments
1. Line 50-51: Another nice paper to cite here: Li, Hong-Tao, et al. "Origin of angiosperms and the puzzle of the Jurassic gap." Nature plants (2019): 1.
2. Figure 1: There is a mismatch between what red and green represent in the caption verses in the legend.
3. Figure 2: The legend should be ‘75%, 50%, 25%’ instead of ‘75%, 50%, 50%’. Same error should also be corrected in Fig S2. Please also adjust the position of label ‘rbcL’ on the upper right circle in Fig 2 to avoid confusion.
4. Line 374: Appendix S16 only contains performance of LSC, SSC, and IR region without considering bootstrap support? Please revise the table or the title of the table accordingly.

Reviewer 2 ·

Basic reporting

1. Line 82: ‘vast majority of studies assume no …. (Wolfe and Rand 2004).’ Is this a review of the studies assuming no conflict among gene trees? If not maybe rephrase this sentence or add in more citations
2. Line 86: Burleigh and Mathews 2007a,b seem to be same citation but slightly different titles?
3. Line 141: ‘using plastomes have excluding non-coding data.’ I think it should be excluded not excluding.
4. Line 164: ‘we refer to this is our ‘accepted tree’, …’ Change is to as.
5. Line 252: ‘the gene was not 1/3rd the length of the amino acid’. I think this could use some rephrasing, it reads like a double negative and a little confusing to understand on the first read.
6. Line 305: ‘The WAG model was not found to be the optimal amino acid for any gene in the analysis,…’ I think the word model is missing after amino acid?
7. Line 329: ‘This pattern is similar to that seen in nuclear phylogenomic datasets (Smith et al., 2015)’. I think the use of a single reference here is a little misleading given the general phrasing of the sentence. There are a lot of other studies that assess gene tree conflict in nuclear datasets that could be cited here, and I would suggest adding more in if appropriate. Is this a review paper that goes over several examples of gene tree conflict in nuclear phylogenomic datasets or is this the only paper specific to angiosperm gene tree conflict?
8. Line 349: ‘We found that a wide range of evolutionary models best fit our data—rather than just a single model for the entire concatenated set of genes’. It is unclear what the authors used to determine best fitting models for their dataset in this instance.
9. Line 374: ‘with both outperforming the IR whether considering bootstrap or not’ I think the word support is missing after bootstrap
10. Line 378,389,392: missing commas in base pair lengths
11. Line 390: ‘In this study, the alignment length of ycf1 was 21,696 bp’ but mentioned in previous sentence that another study found it was ~5,400 bp. Unclear what the discrepancy in length is here.
12. Line 442: ‘Positive relationships between alignment length, tree length and concordance also suggest that stochastic error is at play, because shorter or slower evolving genes harbor less signal to accurately infer relationships.’ But the authors also find that genes that lead to the most concordant trees with the reference are not always the longest, which might suggest systematic error is at play as well. I think that the discussion weighs heavily on the impact of stochastic error on gene tree conflict but only briefly touches on systematic error (and effectively ruling it out in Line 468-472, and the statement in the conclusion about longer genes being superior (line 654-655)). I don’t think from the results that systematic error is convincingly ruled out (see above comments on model performance in the experimental design section), so I would like to see this argument against systematic error toned down a little/caveated a bit more throughout the discussion.
13. Line 524: ‘Unlike in that study, there are reasons to believe that biological gene tree conflict could result in the chloroplast genome…’ Given the broader nature of the reader base, I would suggest talking in a little more detail about what some of the expected biological conflicts between chloroplast gene tree could be.
14. Figure 1: I think it is a little confusing to have green represent two things on one graph (and when zoomed out, green branch looks too similar to black branches on the tree). So I would suggest changing the amino acid alignment color from green to something else.
15. Figure 3: I think some lines pointing from the histogram bar to the relevant gene would be helpful, since the spacing is tight between some of the gene names/ positioning on the histogram, so it is hard to tell what value is relevant for gene.
16. Table 1: It is unclear to me why some many decimal place values are needed in the likelihood scores. I would suggest rounding up to make the numbers more visually palatable.

Experimental design

1. Model influence assessment
I don’t think that rerunning the analysis with only one other model, and only for the amino acid dataset from what I can tell is a particularly strong assessment of model performance/influence (which is still an admittedly hard thing to assess), particularly given the conclusion later on in the paper that the model choice/systematic error isn’t influential in the gene tree conflict described here. Without directly attempting to assess model performance (e.g. Bayesian posterior predictive assessments, some proposed ones in Brown 2014 Detection of implausible phylogenetic inferences using posterior predictive assessment of model fit, Systematic Biology), it would be difficult to say that because two analyses with different models produced the same tree, there has been little to no influence of systematic error, both models could have common assumptions that are violated by the datasets and produce the same wrong tree. I think it would be interesting to more directly test for model performance, but will acknowledge that it can be a computationally demanding task, so would leave that choice to the authors (could also assess influence of alignment method this way as well, which is less computationally demanding than posterior predictive statistics that require simulating posterior distributions of trees). But if they decide not to assess model performance more directly, I would like to see the arguments of minimal influence of systematic error and model influence on the gene tree conflict caveated more in the discussion than it is currently.
2. Model choice
Line 139-141: ‘We did not use non-coding regions; as proper orthology at deep time scales can be difficult to assess due to gene re-arrangements and inversions; this is in part why most deep phylogenetic analyses of angiosperms using plastomes have excluding non-coding data.’
So does this mean that non-coding data was excluded from the nucleotide dataset as well in this study? Was there any model choice analyses applied to the data in which GTR+G was selected as the best fitting model from? I think it would be important to carefully discuss the choice of these models for this dataset. I would think that using coding information only will emphasize the heterogeneity in rates of evolution between regions and/or lineages within the dataset and accommodating more heterogeneity (e.g. site-specific partitioning; a CAT model, etc) into modeling might be more suitable/appropriate for phylogenetic inferences, so I think readers concerned about model performance with the dataset would appreciate careful discussion of model choices, as well as the choices to use the same model across all gene regions despite what I think is an acknowledgment that there seems to a range of models that fit across the data set rather than a single one (line 349)?

3. Gene tree conflict assessment
I think it would be a nice addition to the study to characterize just how much conflict exists among individual genes about the angiosperm relationships directly. There are a lot of sentences that mention conflict is prevalent (e.g. Line 422: ‘We would also expect all plastid genes to show similar patterns of conflict when compared to non-plastid inferred phylogenies’), but I don’t think it is ever directly assessed this in the dataset. One way I think it would be easy to do so would be to do some pairwise comparison of concordance among the gene trees themselves, rather than just to the reference tree. It might also be interesting to see if the longer genes agree with each other, in addition to the comparison of whether they agree with the reference trees that is already presented in this study. It might be able to give some assessment of accuracy that independent of assuming that the reference tree is correct about the relationships.

4. Reference/plastome tree methodology
I think the authors should explain in a bit more detail about why they chose to use a concatenation method when inferring plastome-wide trees, especially since it seems like in the discussion they warn readers away from doing that. Why not make a ‘plastome’ tree using a species tree method given that there is a lot of gene tree variation going on in the data set, of which the species tree methods attempt to control more for than concatenation methods? Were concatenation methods used to match the current angiosperm phylogenies because the reference trees pulled from other studies were inferred from concatenation methods or as an example of common methods used in studies aiming to determine relationships among angiosperms to highlight the problems with using them when lots of gene tree variation exists in a dataset? I think a little more detail on the choice of phylogenetic inference methodologies used in this study will provide some need clarity.
5. Choice of Binomial regression
I think the section explaining the generalized linear model is a little confusing and could use some clarity. For example, it is a little unclear to me what format the response variable (concordance) was in, what it actually binomial (concordant vs not over individual nodes) or proportional (percent concordance across entire tree), and was the model run per gene or across all genes? If it was the proportional data, is the quasibinomial the best family to use? Why not negative binomial, which is also used to account for overdispersion but is more traditionally used with proportional data? If it was actually binomial data, was a model for each individual gene and if so, where are these reported? If one model was run over all the genes with binominal data, is gene incorporated as a random effect in the model? I also wonder if the authors used the binomial data and could treat gene as a random effect, would they be able see more subtle patterns by being able to add gene as a random factor to help account for variance that might not be explained by the chosen predictors (e.g. alignment length). I will admit that I have not played around with these particular models with binary data myself all that much, so please take the latter comments about using the binomial data itself as mere suggestion of something to look into and not strong request. But either way, I think the description of this methodology could use some further clarity.
Negative binomial (distributions similar to poison
6. Line 183: Why the choice of 70% bootstrap support for moderate support nodes? Is not 50% is a more common cutoff (e.g. majority rule consensus trees)? Or wouldn’t we want bootstrap support to be near 100% for nodes that are informative about relationships? I would suggest including a sentence with some justification for bootstrap cut off values for nodes that are informative about relationships vs not informative
7. Line 205 says the authors calculated a set of summary statistics for each gene, including the number of included species, but it is unclear to why some species would be included or excluded for a particular gene.
8. Line 234 ‘Investigation of model fits on full datasets revealed….’ It is unclear what is being used to investigate model fit on full datasets in this context, it this assessment significance with logistic regression/quasibinomial regression?
9. Line 247: ‘We also performed saturation analyses…’ I am not sure the average reader will be familiar with this analysis so I would suggest explaining what it is and what it can tell you in a little more detail.
10. Line 265: “as described above for each of the ‘plastome’ analyses”. This wording is a little vague and it’s unclear to me what the plastome analyses was. I would suggest rephrasing this sentence for clarity.
11. Line 314: ‘Quasibinomial regression results for amino acids aligned with FSA and cleaned for 30% minimum occupancy were not significant for all predictors, but this model had an abnormally high estimated dispersion, suggesting this finding to be in error’. I think this sentence could use a little more clarity, it is unclear what the error is here, the alternative alignment method tests having no influence on the results?

Validity of the findings

1. Line 106: ‘Our results show that the plastome itself tends to perform well at all levels; however, it contains notable gene tree conflict.’ Is it performing well if there is notable gene tree conflict? I would suggest clarifying what is meant by the plastome and good performance?
2. Line 107-109: ‘The number of conflicting genes at each node is often comparable to the number of concordant genes; however, the majority of plastid genes are uninformative for most nodes when considering support and thus unlikely to positively mislead studies.’ (Also something similar is mentioned again on lines 641-643). I think that it may be a little more nuanced than this and worthy of stating in more detail in this discussion. While any one gene region that is uninformative on its own about phylogenetic history won’t mislead a study (because it doesn’t have much to say in the first place), it is possible that a study using a large set of genes to infer a phylogeny, one with a lot of less informative regions and only a few high-information regions can be really sensitive to things like poor model performance/systematic error for those few highly informative regions and end up with strong support for an incorrect phylogeny, especially when using methods that don’t try to account for variation in gene tree information content (like concatenation methods). I think that this is a way in which having a dataset with lots of less informative regions in it could mislead studies and a reason why there has recently been more effort to explore how information content is distributed across genes in our phylogenomic datasets. This is explored more in a few studies (e.g. Thomson and Brown 2017 Bayes factors unmask highly variable information content, bias, and extreme influence in phylogenomic analyses, Systematic Biology and Shen et al 2017 Contentious relationships in phylogenetic studies can be driven by a handful of genes, Nature Ecology and Evolution) and I would suggest maybe discussing this area in more depth when talking about the impact of lots of less informative regions in phylogenomic datasets, perhaps in the discussion.
3. Lines 112-115: ‘However, some nodes across angiosperm phylogeny show at least several strongly supported conflicting genes, this rare event indicates an area for future exploration, as it provides evidence that the chloroplast is potentially affected by a biological source of gene-tree conflict …’ Could it not also be stemming from systematic error (e.g. poor model fit)?
4. Line 119: ‘Although as a whole the chloroplast performs well in topology reconstruction, ...’ Again, I think it would be helpful to clearly lay out what is meant by performing well. Agreeing with known reference trees?

Additional comments

This is a solid initial investigation into the sources of gene tree conflict potentially underlying strong conflict among angiosperm phylogenies. I think this paper adds to the growing number of phylogenomic studies that are trying to more rigorously investigate methodological choices that can lead to particular inferences about relationships among organisms. I find the results compelling that there is extensive variation and that the paper has useful considerations and suggestions for researchers interested in inferring a robust angiosperm phylogeny.

The paper presents good evidence that by choosing particular gene regions to minimize the impact of stochastic errors, researchers can produce more robust angiosperm topologies (or at least ones that are more concordant with the available phylogenies used in this study as reference trees). However, I think multiple studies have shown the impact of systematic error in producing strong conflicts that aren’t necessarily related to the level of phylogenetic signal/information content and that the arguments presented in this paper about the minimal influence of systematic error aren’t strongly supported by the analyses/results presented in this study. I realize thorough investigations of model performance are very computationally burdensome and the ability of available test statistics to accurately assessing model performance is not all that clear yet, so a thorough investigation of model performance in this study would be a high bar request for sure, and I am not suggesting that the evidence presented is not any good for leading readers to more robust phylogenetic inference from chloroplast genomes – just that perhaps the strength of claims in some sections of the paper need to be tempered a bit more with caveats. I would also suggest adding on to the concordance analyses the pairwise comparison of concordance among the gene trees themselves to highlight the extent of gene tree conflict across the plastome and more clarity in the descriptions of several of the methodological choices of this study (listed in more detail in the other sections).

Reviewer 3 ·

Basic reporting

Very well done. I find no noticeable issues, so "no comment".

Experimental design

No comment.

Validity of the findings

No comment.

Additional comments

Walter et al. have presented a very nice study demonstrating variation in phylogenetic signal across different genes in the plastome. I cannot see any issues with the paper from a technical or presentation standpoint. I very much appreciate that the authors state things very matter-of-the-factly and do not push the capacity of their findings. As a person that does a lot of whole chloroplast sequencing, this will make me consider the different impacts of portions of my alignments on my final phylogeny, especially when doing divergence time estimation.

I found a few minor grammatical/content issues:

Line: 61-62: Plastomes are not always inherited paternally in gymnosperms. It depends on the species.

Line 135: “utilized” should be used”.

Line 263: The literature usually refers to these as Large Single Copy and Small Single Copy, though there are instances of Long and Short.

Line 299: Give RAxML version.

Line 300: “Used” not “utilized”.

---

## Round 0.2 · accepted · Accept

It looks like the reviewers are very pleased with your revisions. Make sure to thank them if you haven't already (in proof should be fine).

-- Scott

·

Basic reporting

All comments have been appropriately addressed.

Experimental design

No comment.

Validity of the findings

No comment.

Additional comments

Good job!

Reviewer 2 ·

Basic reporting

I am satisfied with the corrections made by the authors in this revised manuscript!

Experimental design

The added explanations of methodological choices in this revised version improved the clarity in the manuscript.

Validity of the findings

No comment

Additional comments

I think the authors did a good job taking all three reviewers & AE's comments into consideration and the resulting manuscript is much improved.